# Role of Magnetic Nanoparticles Size and Concentration on Structural Changes and Corresponding Magneto-Optical Behavior of Nematic Liquid Crystals

**DOI:** 10.3390/nano12142463

**Published:** 2022-07-18

**Authors:** Peter Bury, Marek Veveričík, František Černobila, Natália Tomašovičová, Katarína Zakuťanská, Peter Kopčanský, Milan Timko, Markéta Jarošová

**Affiliations:** 1Department of Physics, FEIT, Žilina University, Univerzitná 1, 01026 Žilina, Slovakia; vevericik@fyzika.uniza.sk (M.V.); fero@fyzika.uniza.sk (F.Č.); 2Institute of Experimental Physics, Slovak Academy of Sciences, Watsonova 47, 04001 Košice, Slovakia; nhudak@saske.sk (N.T.); zakutanska@saske.sk (K.Z.); kopcan@saske.sk (P.K.); timko@saske.sk (M.T.); 3Institute of Physics of Czech Academy of Sciences, Cukrovarnícka 10, 162 00 Praha 6, Czech Republic; jarosova@fzu.cz

**Keywords:** ferronematics, spherical nanoparticles, magneto-optical properties

## Abstract

The effect of magnetic nanoparticles size and concentration on nematic liquid crystal (NLC) behavior in a magnetic field was investigated. The magneto-optical investigation using measurements of the light transmission through the liquid crystal was used to study the structural changes induced by an applied weak magnetic field. Magnetic nanoparticles Fe_3_O_4_ of spherical shape with different size and volume concentration were added to NLC 4-cyano-40 -hexylbiphenyl (6CB) during its isotropic phase. In contrast to undoped liquid crystals, the distinctive different light transmission responses induced by a magnetic field in studied NLC samples were observed suggesting both structural changes and the orientational coupling between magnetic moments of nanoparticles and the director of the NLC. Experimental measurements were conducted, including investigation under linearly increasing and/or jumped magnetic field, respectively, as well as the investigation of time influence on structural changes to study their stability and switching time. The analysis of observed light transmission characteristics confirmed the role of concentration and size of magnetic nanoparticles on the resultant behavior of investigated NLC compounds. The obtained results showed the lowering of the threshold magnetic field with an increase in the volume concentration of nanoparticles and on the important role of nanoparticles size on stability and switching properties. Obtained results are discussed within the context of previous ones.

## 1. Introduction

The liquid crystal-based nanomaterials that can be ordered or reoriented by external stimuli such as magnetic and electric fields, temperature or light can be very attractive from the designing point of view. In addition, combination of the anisotropy of liquid crystals (LCs) with the unique properties of nanoparticles creates new perspective opportunities for regulating the LCs properties and, in doing so, produces new possibilities for materials which can make a proposal of the nanotechnology development in areas such as display devices, nanoscale electronics, sensors or electro-optics [1,2,3]. One of such composites are nematic liquid crystals (NLCs) doped with magnetic nanoparticles (ferronematics), with magnetically active anisotropic fluids which represent an original idea [4], indicating that doping of LCs by adequate magnetic particles can improve their sensitivity to magnetic fields, and was consequently experimentally verified [5,6,7,8,9]. The basic particularity of ferronematics is a coupling between magnetic particles and NLC molecules. The coupling between director ***n*** characterizing the preferred configuration of the nematic molecules and average magnetic moment ***m*** of magnetic particles can result under an applied magnetic field in the reorientation of NLC molecules [6,7,10,11,12]. Magnetic particles are in principle affected by an external magnetic field and sequentially, due to the interaction between magnetic particles and NLC molecules, such composites could be interesting in terms of practical application. In order to ensure that the inter-particle magnetic interaction can be ignored, the small volume concentration of magnetic particles should be used. As a consequence of improving sensitivity to magnetic field, a small concentration of magnetic nanoparticles is actually added into NLC and has an effect on its other behaviors, including electro-optical and magneto-optical properties, threshold field or phase transition temperature [13,14,15,16,17,18,19], etc.

As theoretical works have assumed [20], magnetic nanoparticles of spherical shape reduce temperature of isotropic-nematic phase transition, whereas anisotropic nanoparticles increase this temperature. Results of experimental works [17,21] really showed that doping with spherical iron oxide nanoparticles leads to lowering of isotropic-nematic phase transition temperature, while particles in the nanorods shape causes a shift to higher values, however, the effect increases with an increase in concentration. The organic surfactant, which prevents aggregation, can cause another contribution to the shift of the isotropic to nematic phase transition temperature, with increasing particle concentration. Many experimental results have been reported [18,22,23,24] and theoretical studies performed [25,26,27] about the decrease in threshold magnetic field observed in NLCs doped with magnetic nanoparticles, however, it was shown that an increase in the threshold of magnetic fields can also occur [28,29]. As such, the influence of nanoparticles on NLC compound properties depends on their particularities, which are related to their material, size, shape, concentration, etc.

The addition of magnetic particles to NLCs, except for the reduction in the threshold magnetic field or shift in the isotropic-nematic phase transition temperature, can also significantly change, and mostly improve, optical properties of the originally pure NLC, including their switching behavior [30,31,32]. The magneto-optical effect was investigated in lyotropic LCs doped with ferrofluid in their isotropic phase [33,34] and in NLCs doped with magnetic particles [35,36,37,38]. The redistribution of the nematic director from the uniform alignment, due to the magnetic field variation, can be also accompanied by light intensity oscillations [39,40]. The obtained results also showed that the optical activity on NLCs compounds affected by doping with magnetic particles results in the elliptical polarization of the emerging linearly polarized light beam under an increasing magnetic field caused by the distortions of the nematic order by magnetic particles.

The present paper by a certain measure follows the articles [21,41], in which the effect of nanoparticle size and concentration on isotropic to nematic phase transition temperature and threshold magnetic field of 6CB liquid crystal doped with spherical iron oxide nanoparticles was investigated using capacitance and DSC measurements. It was found that the transition temperature from isotropic to nematic phase decreases for all the composites comparing to pure liquid crystal and for the composites containing large concentration of nanoparticles dramatic changes even occurred. The obtained results also showed the threshold magnetic field decreases with an increase in the volume concentration of nanoparticles. Estimated calculations based on results obtained from experimental measurements and theoretical predictions [41] suggested soft anchoring between the nanoparticles magnetization vector and liquid crystal director. The aim of the paper is to make clear the role of nanoparticle size and the concentration of spherical iron oxide nanoparticles on NLC optical properties, with the intention of controlling the properties of the primal liquid crystal. In this contribution, we report results of magneto-optical behavior using light transmission measurements on the set of NLC (6CB) doped with spherical magnetic nanoparticles of different size and concentration to reveal their optical properties in respect to the structural changes that occur under a magnetic field.

## 2. Experimental

The iron oxide nanoparticles with diameters 5 nm, 10 nm, 15 nm, and 20 nm coated with oleic acid dispersed in chloroform were purchased from Ocean Nanotech (San Diego, CA, USA). The composites of thermotropic NLC 4-cyano-40 -hexylbiphenyl (6CB) and iron oxide nanoparticles were prepared by the following procedure. Nanoparticles originally dispersed in chloroform were mingled with NLC in its isotropic phase. Consequently, the mixture of NLC and nanoparticles was stirred in the isotropic phase while the chloroform was completely evaporated. By this procedure, the composite with volume concentration 10^−3^ was prepared. Samples with other concentrations, i.e., with volume concentrations 5 × 10^−4^ and 10^−4^, were acquired by admixing an additional amount of NLC. The same procedure was repeated for nanoparticles with diameters 5 nm, 10 nm, 15 nm, and 20 nm leaving twelve ferronematic samples—three volume concentrations for each of three nanoparticle sizes. The real sizes of Fe_3_O_4_ nanoparticles were determined from transmission electron microscopy (TEM) images (Figure 1). TEM measurements, which give sizes of not coated iron oxide grains, that means without oleic acid layers, were performed on transmission electron microscope FEI Tecnai G2 20 (FEI, Hillsboro, OR, USA) with a LaB_6_ cathode operating at acceleration voltage 200 kV. The microscope is equipped with a CCD camera, Olympus Veleta (Tokyo, Japan). Corresponding results for all four groups of the used nanoparticles are presented in Figure 1. The obtained results show very uniform size distribution of nanoparticles. Concerning the size distribution, while the size distributions for pure Fe_3_O_4_ nanoparticles are narrow, the size distributions of the Fe_3_O_4_ nanoparticles with oleic acid coating are wider. Therefore, oleic acid coating is probably not uniform for all nanoparticles.

To find the effect of spherical magnetic nanoparticles on nematic liquid crystal, 6CB optical properties light transmission investigations were conducted in LC cells with a 50 μm cell gap and side glasses coated with ITO transparent conductive layers and alignment layers rubbed in a parallel direction to the electrodes, like in the case of capacitance measurements [41]. The laser beam (532 nm, 5 mW) generated by Green DPPS Laser Module CW532 (Roithner LaserTechnik GmbH, Wien, Austria) illuminated the cell’s glass in normal incidence formerly passing an optical triangular prism and polarizer 1. The initial position of polarizer 1, for all investigated compounds, ensured the linearly polarized incident light beam, the NLCs position provided for maximal transmittance registered by photodetector. The polarizer 2 could be adjusted to obtain maximal transmittance after the partial depolarization caused by NLC (parallel arrangement) or minimal transmittance (crossed arrangement). After passing polarizer 2 and prism, the intensity of transmitted light was recorded by a photodetector (Thor LABS PDA36A 350–1100 nm) connected to the multimeter and subsequently registered by computer monitoring the light transmission as a function of magnetic field or time. The scheme of experimental arrangement is presented in Figure 2. The light transmission was then expressed in the case of parallel polarizers as *I/I*_0_ and as (*I* − *I*_0_)/*I*_0_ in the case of crossed polarizers, where *I*_0_ and *I* are the intensity of incident light passing through the LC cell without an applied field and under the field, respectively.

## 3. Results and Discussion

Magneto-optical behavior was experimentally investigated using light transmission measurements in the sets of NLCs doped with spherical magnetic nanoparticles of different size and concentration using the experimental arrangement shown in Figure 2 and described in the previous section. The light intensity as a function of magnetic field or time was registered and the light transmission calculated. Presented light transmission characteristics were also compared with characteristics obtained using crossed polarizers that showed the similar but opposite behaviors, however, the light intensity changes were weaker but oscillations more pronounced.

Figure 3a–d shows the dependence of light transmission on a magnetic field for 6CB nematic liquid crystal doped with spherical nanoparticles of all different sizes (5 nm, 10 nm, 15 nm, and 20 nm) and volume concentrations (10^−3^, 5 × 10^−4^ and 10^−4^). Presented dependences confirmed the marked effect of nano of all sizes and concentrations on the light transmission in 6CB. However, the difference in behavior of individual compounds comparing to pure 6CB (Figure 3a–d) depends on the concrete size and/or concentration of nano. The obtained results approved the role of both size and concentration on the magnetic threshold field, but also (of no less importance) on the process of its further development. The threshold field, defined as the field where the light transmission is decreased to 90% of the initial value (zero field) [36], decreases with increasing concentration only slightly, except the more rapid decrease registered in the case of highest concentration (10^−3^) but only in the case of compounds with the smallest (5 nm) and highest (20 nm) sizes. The position of the threshold field, however, is influenced by the presence of ferroparticles and it is attributed to the combination of both ferromagnetic and anchoring energies, as it depends on both the concentration and size of nanoparticles.

Concerning two lower concentrations (5 × 10^−4^ and 10^−4^), only negligibly small changes in the threshold field for these compounds compared to pure 6CB were found in contrast with some compounds with volume concentration 10^−3^. The preferred parallel orientation of magnetization vector to liquid crystal director suggests that the threshold magnetic field should be shifted to lower values. Curves for composites with volume concentration 10^−3^ have threshold fields more or less pronounced, markedly for a composite containing 5 nm and 20 nm particles. The shifts for composites with nanoparticle volume concentrations 10^−4^ and 5 × 10^−4^ are so close that they can be distinguished only with difficulties. Using the previous capacitance experiments, the surface density *W* of the anchoring energy of the NLC, with the spherical nanoparticles and parameter ω characterizing anchoring, were calculated resulting in the soft anchoring was determined for these kinds of NLC compounds [41]. This fact could be the reason for some ambiguous behavior. It is also interesting that when comparing results obtained for 10 nm and 15 nm, the more stable dependences were registered for a smaller size of nanoparticle. Another reason could be that oleic acid was used as surfactant, for which the coating in the case of all kinds of such small nano is not uniform.

The light transmission in the case of highest concentration (10^−3^) and largest spherical nano (20 nm) begins to change just after the magnetic field is applied followed by the increase in the light transmission over the normalized value. Such development can be attributed to the presence of different regimes of the threshold orientational behavior of compensated ferronematics under a magnetic field [11,27,37]. The reason for that may be due to the gradual magnetization of the ferronematic in the direction of the director, rotation of the director in the external field, and finally synchronous rotation of the director and magnetization of compensated ferronematics along the applied field. Similar development of magneto-optical characteristics, except for the increasing part, was also observed in the case of 6CHBT doped with functionalized SWCNTs of the same concentration [32] and/or rod-like magnetic nanoparticles [39].

Another reason for this behavior, however, could be caused by the processes of aggregation that are represented in the cases of higher concentration by the actual lowering of effective nanoparticle concentration. Concerning the nematic-isotropic transition temperature, *T_NI_*, the shift toward lower temperatures was registered for all investigated composites [42]. As regards to oscillations, it is assumed that when the magnetic field is passing through a threshold field, a planar aligned molecular director undergoes a reorientation between the initial planar and final perpendicular orientation. Therefore, when the NLC is subjected to a laser beam, a succession of maxima and minima of the transmitted light through the NLC cell appears [7]. The same situation occurs when the magnetic field is decreased and the NLC returns to the initial planar orientation. The same oscillations are present also on the curve corresponding to the pure NLC that corresponds to the fact that they are the peculiarity of NLC molecules, which are under a magnetic field and illumination, so that the transmission oscillations are accordingly induced by a reorientation of the molecular director when the NLC is subjected to an external field higher than the transition threshold and to a laser beam.

Figure 4 presents the mutual reciprocal comparison of light transmission characteristics measured under a magnetic field for 6CB doped with spherical magnetic nanoparticles with size 5 nm, 10 nm, 15 nm and 20 nm, including pure 6CB, for two different concentrations; lowest (10^−4^) and highest (10^−3^). While the light transmission dependences for the compounds with lowest concentration (Figure 4a) are characteristic by superimposed oscillations on detected curves registered after the threshold field transition and there is only a very slight difference between individual threshold fields, in the case of highest concentration (Figure 4b) oscillations are registered only for the smallest nano size (5 nm). The anomalous shift in the threshold fields in the case of highest concentration can be affected by the different level of aggregation, at which the different coating by surfactant could play an important role, too.

Observed light transmission responses pointed out that the decrease in the oscillation amplitude, with even oscillations vanishing, and by that the higher stability of light transmission responses can be ensured by an increase in both the concentration and size of the magnetic nanoparticles in NLC. As regards to peculiar behavior of the compound with the highest concentration, this has already been described previously.

The comparison of measurements for both increasing and decreasing magnetic field is illustrated in Figure 5 for two different concentrations and two different sizes of nanoparticles. This shows quite a clear hysteresis in the region of decreasing light transmission after passing the threshold field Similar hysteresis was also obtained for other particle concentrations and sizes. However, no, or only very weak memory effects were observed in light transmission characteristics, and it is evident that the light transmission at decreasing magnetic field directs toward or very close to the initial value. The observed hysteresis also indicates that at higher fields some process of magnetic nanoparticles aggregation can occur.

The progress of switching processes including the comparison of effects of external magnetic field strength (200, 300 and 400 mT) on the light transmission for 6CB samples doped with spherical magnetic nanoparticles of 10 nm size and concentrations 10^−4^ (a), 5 × 10^−4^ (b) and 10^−3^ (c), as representative compound, after jumped magnetic field changes is illustrated in Figure 6. Observed developments apparently validate that doping with spherical nano can markedly influence the reorientation of NLC molecules, in the case of an applied magnetic field. However, the relaxation time depends on the strength of the magnetic field and concentration or size of nanoparticles. The relaxation time of the processes occurring after the magnetic field is applied is noticeably shorter than the relaxation time of processes after removing the magnetic field, which are almost the same for all used magnetic field intensities. The behavior, when the rate of processes occurring after the application of external field depends on the strength of applied fields, coincides with previous electro-optical [43], but also magneto-optical [32,39] investigations. Observed responses are fully consistent with the development of light transmission characteristics observed in an increasing magnetic field, as presented in Figure 3. While the time responses of NLC compounds doped with lower concentrations (10^−4^) have some small indication of superimposed oscillation, no visible oscillations were registered for higher concentrations. The increase in the concentration, in addition, ensures the stability of switching developments. It is evident that the anchoring in the case of spherical nanoparticles can ensure (for their higher concentrations and larger sizes) the stable magneto-optical responses.

This fact can coincide with previous results presupposing the role of a process of nanoparticles aggregation [32,37]. Oscillations occurring after the magnetic field was switched on or switched off, were observed in NLCs doped with carbon nanotubes [44], and also when the electric field was applied [30]. The reason for these oscillations can be imputed to the redistribution of NLC molecules direction around the equilibrium position of the nematic director, changing from planar alignment on the cell surfaces to the perpendicular one at the center of the NLC layer, when the magnetic field increases rapidly [33]. The shape of oscillations of a saturation regime observed during measurements of the phase difference in magnetically doped NLCs were taken as a part of their collective behavior [45].

Figure 7 presents the comparison of the light transmission time responses of 6CB doped with spherical magnetic nanoparticles of different sizes (5, 10, 15 and 20 nm) after magnetic field 400 mT pulses were applied for two different concentrations, 10^−4^ and 10^−3^. It is evident that the stability of light transmission response is a strong function of both concentration and particle size. However, the interesting result following the analysis of all responses is that the most stable and quick response is registered for compounds with 10 nm particle size and definitely the highest concentration.

## 4. Conclusions

In this contribution, the role of spherical magnetic nanoparticles size and concentration on the magneto-optical behavior of the nematic liquid crystal 6CB was investigated using light transmission measurements. The light transmissions were examined utilizing an initially linearly polarized laser beam passed through liquid crystal cell under a continuously increasing and/or jumped magnetic field. The obtained results confirmed both previously supposed orientational coupling between magnetic nanoparticles and the liquid crystal matrix and their important role on structural changes under a magnetic field, resulting in the pronounced improvement of optical properties, including the switching behavior. Except for an increase in the threshold field and stronger magneto-optical responses caused by nanoparticle size and concentration, it was possible to achieve higher magneto-optical stability by decreasing the superimposed oscillations amplitude. In addition, investigations pointed out that the surfactant coating can play an important role. Presented magneto-optical measurements also supported the conclusion that the soft anchoring is above all the characteristic property of spherical magnetic nanoparticles in NLC suspension.

## Figures and Tables

**Figure 1 nanomaterials-12-02463-f001:**
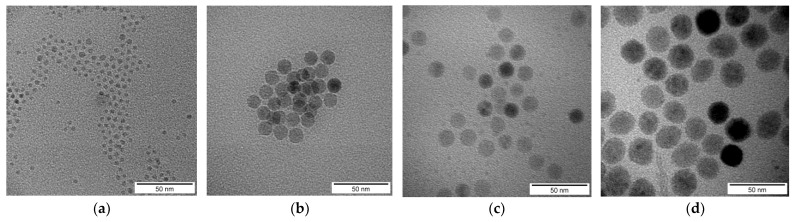
TEM image of 5 nm (**a**), 10 nm (**b**), 15 nm (**c**) and 20 nm (**d**) Fe_3_O_4_ nanoparticles.

**Figure 2 nanomaterials-12-02463-f002:**
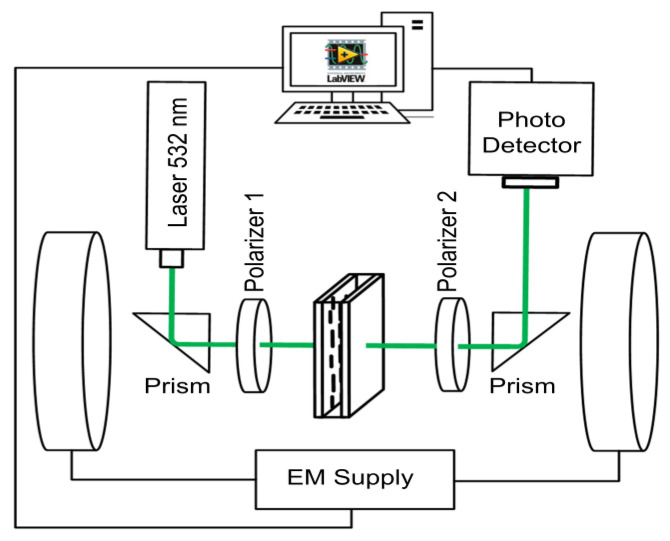
Experimental arrangement for magneto-optical investigation.

**Figure 3 nanomaterials-12-02463-f003:**
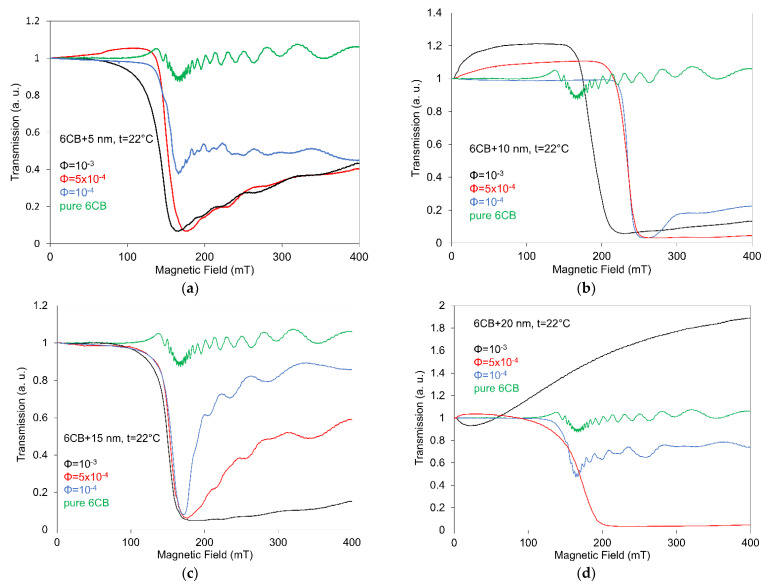
Dependences of light transmission on magnetic field for 6CB liquid crystal doped with spherical magnetic nanoparticles with size 5 nm (**a**), 10 nm (**b**), 15 nm (**c**) and 20 nm (**d**) each of concentrations 10^−4^, 5 × 10^−4^ and 10^−3^, including pure 6CB.

**Figure 4 nanomaterials-12-02463-f004:**
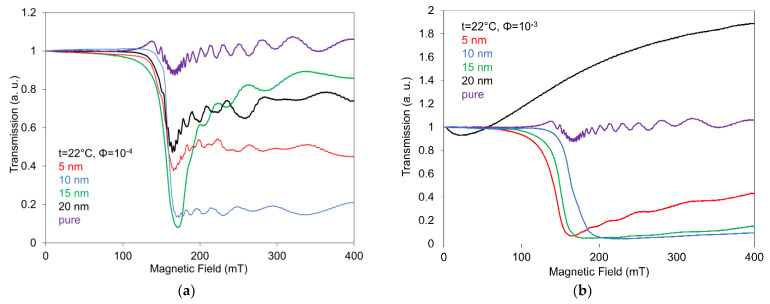
Dependences of light transmission on a magnetic field for 6CB liquid crystal doped with spherical magnetic nanoparticles with size 5 nm, 10 nm, 15 nm and 20 nm for the concentrations 10^−4^ (**a**) and concentration 10^−3^ (**b**), including pure 6CB.

**Figure 5 nanomaterials-12-02463-f005:**
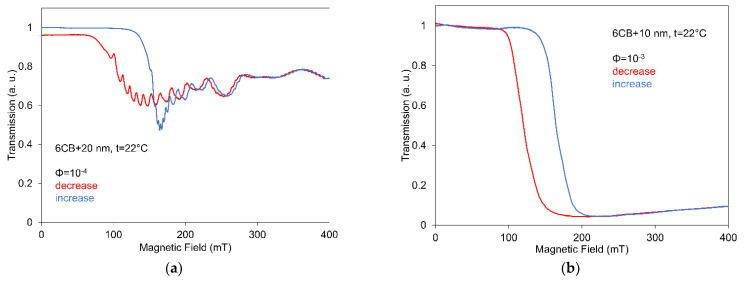
Magneto-optical characteristics of 6CB liquid crystal doped with spherical magnetic nanoparticles with volume concentrations 10^−4^ (**a**) and 10^−3^ (**b**) measured for increasing and decreasing magnetic field.

**Figure 6 nanomaterials-12-02463-f006:**
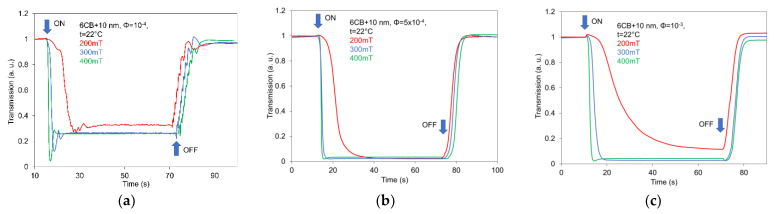
Light transmission time responses registered on 6CB liquid crystal doped with spherical magnetic nanoparticles of 10 nm size and different concentration 10^−4^ (**a**), 5 × 10^−4^ (**b**), and 10^−3^ (**c**) after jumped magnetic field changes of 200 mT, 300 mT, and 400 mT were applied.

**Figure 7 nanomaterials-12-02463-f007:**
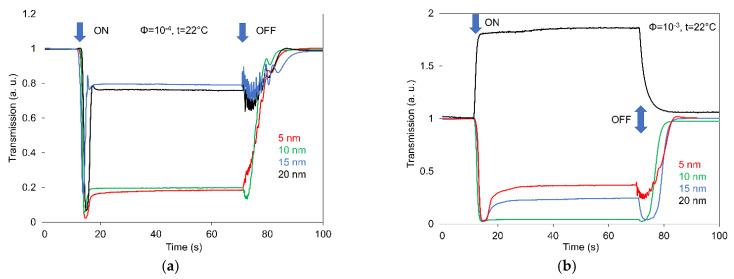
Comparing of light transmission time responses of 6CB doped with spherical magnetic nanoparticles of different sizes (5, 10,15 and 20 nm) after magnetic field 400 mT pulses were applied for two different concentrations, 10^−4^ (**a**) and 10^−3^ (**b**).

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
