# Peer review of "Role of Magnetic Nanoparticles Size and Concentration on Structural Changes and Corresponding Magneto-Optical Behavior of Nematic Liquid Crystals"

_nanomaterials, 2022, doi:10.3390/nano12142463_

Round 1
Reviewer 1 Report
The manuscript presents an interesting study of magneto-optic effects in liquid crystals composites of liquid crystals with iron-oxide nanoparticle. The manuscript is highly experimental and presents valuable results of static and dynamic behaviour of these composites in magnetic field. The overall opinion about the article is good and it can be considered for publication but there are some minor presentation aspects that should be fixed.
· In Fig.1 there is no clear information about the images presented there. The images were taken on coated nanoparticles? “Figure 1a and 1b are identical- but there are 2 nanoparticle’s size mentioned there.
· “LC cells with a 50 μm cell gap and side glasses” is not clear, what is the side glasses ?
· Line 150: both sizes I believe you mean all sizes because there are more than 2
· The curves obtained in Fig.3 to Fig5 were recorded on a continuous variation of the magnetic field or there were discrete recordings and the lines are just guides to the eyes? In this case I suggest you put the plots there instead of a continuous line
· How did you keep the temperature of the sample constant?
· Fig.4 a, I believe the pure 6CB correspond to the purple line not the black one
· In Fig. the authors mention two different concentrations and two different sizes but there is no comparison between the same size of two concentrations. I believe thus the influence of the concentration can be clearly spotted.
Author Response
Dear reviewer
I accepted all your remarks.

Reviewer 2 Report
The manuscript entitled " Role of Magnetic Nanoparticles Size and Concentration on Structural Changes and Corresponding Magneto-optical Behavior of Nematic Liquid Crystals" written by Peter Bury, Marek Veveričík et.al reported that The effect of magnetic nanoparticles size and concentration on nematic liquid crystal (NLC) behavior in magnetic field was investigated. The magneto-optical investigation using measurements of the light transmission through the liquid crystal was used to study the structural changes induced by applied weak magnetic field. Magnetic nanoparticles Fe3O4 of spherical shape with different size and volume concentration were added to NLC 4-cyano-40 -hexylbiphenyl (6CB) during its isotropic phase. In contrast to undoped liquid crystals the distinctive different light transmission responses induced by magnetic field in studied NLC samples were observed suggesting both structural changes and the orientational coupling between magnetic moments of nanoparticles and the director of the NLC.
In my view, there are some questions need to be solved before it can be accepted for publication. Here is the detail of necessary revision,
1. Reagents mentioned in the manuscript should indicate the purity.
2. The authors should rearrange the references, citing more references from the last 5 years.
3. The two pictures in Figure 1 (a,b) in the manuscript are exactly the same. Please correct.
4. It is necessary for the author to modify Figure 2 to make the drawing style fit the article.
5. The legend style of Fig. 6(c) is different from that of Fig. 6(a,b). Please revise to be consistent.
Author Response
Dear Reviewer,
we accepted all your remarks.

Round 2
Reviewer 2 Report
Accept in present form.